# Investigating the role of stratospheric ozone as a driver of inter-model spread in $CO_2$ effective radiative forcing.

Rachael E. Byrom[1], Gunnar Myhre[1], Øivind Hodnebrog[1], Dirk Olivié[2], Michael Schulz[2]

[1]CICERO, Oslo, 0318, Norway

[2]Norwegian Meteorological Institute, Oslo, 0313, Norway

*Correspondence to*: Rachael E. Byrom (rachael.byrom@cicero.oslo.no)

**Abstract.** Addressing the cause of inter-model spread in carbon dioxide ($CO_2$) radiative forcing is essential for reducing uncertainty in estimates of climate sensitivity. Recent studies demonstrate that a large proportion of this spread arises from variance in model base-state climatology, particularly the specification of stratospheric temperature, which itself plays a

dominant role in determining the magnitude of $CO_2$ forcing. Here we investigate stratospheric ozone ($O_3$) as a cause of inter-model differences in stratospheric temperature, and hence its role as a contributing factor to spread in $CO_2$ radiative forcing. We use the Norwegian Earth System Model 2 (NorESM2) to analyse the impact of systematic increases/decreases in stratospheric $O_3$ on the magnitude of $4xCO_2$ effective radiative forcing (ERF) and its components. Firstly, we demonstrate that accurate estimation of instantaneous radiative forcing (IRF) requires the use of host-model radiative transfer calculations.

Secondly, we show that a 50% increase and decrease in stratospheric $O_3$ concentration leads to significant differences in base-state stratospheric temperature, ranging from +6 K to -9 K, respectively. These experiments impact IRF primarily due to the influence of base-state stratospheric temperature on the emission of outgoing longwave radiation, with the spectral overlap of $CO_2$ and $O_3$ playing a subsidiary role. However, the impact on IRF does not result in a correspondingly large spread in $CO_2$ ERF. We conclude that inter-model differences in stratospheric $O_3$ concentration are therefore not predominantly

responsible for inter-model spread in $CO_2$ ERF.

## 1 Introduction

Effective radiative forcing (ERF) quantifies the top-of-atmosphere (TOA) perturbation to the Earth's energy balance imposed by a forcing mechanism, such as $CO_2$, aerosols or solar irradiance. It includes the instantaneous radiative forcing (IRF; i.e., the initial radiative response to the perturbation) and the subsequent radiative effect of adjustments in tropospheric and

stratospheric temperature, water vapour, surface albedo and clouds, which each cause an impact on TOA radiative fluxes (Myhre et al., 2013; Boucher et al., 2013; Sherwood et al., 2015; Forster et al., 2021).

ERF can be expressed simply (following e.g., Chung and Soden 2015a; Smith et al., 2018) as:

$$ERF = IRF + A_{T_{Strat}} + A_{T_{Trop}} + A_{H_2O} + A_\alpha + A_c + \epsilon, \tag{1}$$


whereby ERF is the net (shortwave plus longwave) change in downward TOA flux (W m$^{-2}$), IRF is the direct net change in downward TOA flux (W m$^{-2}$), $A_x$ is the radiative adjustment from stratospheric temperature ($T_{Strat}$), tropospheric temperature ($T_{Trop}$), water vapour ($H_2O$), surface albedo ($\alpha$) and clouds (c), with $\epsilon$ representing a non-linear residual term that is typically small (around 10% of the ERF; Shell et al., 2008).


ERF is used extensively to compare the relative strength of different forcing agents. Historically, quantifying the climate impact of a given agent commonly relied solely on diagnosing its IRF or stratospheric temperature adjusted radiative forcing (SARF, e.g., Ramaswamy et al., 2019). However, given that additional so-called 'adjustments' develop from the initial radiative perturbation and impact the TOA imbalance, it is also necessary to include them in the radiative forcing framework.

Consequently, this has been shown to improve the utility of the radiative forcing metric in predicting global-mean surface temperature change ($\Delta T_s$), ultimately due to a more realistic separation of forcing from surface-temperature driven feedbacks (e.g. Sherwood et al., 2015; Marvel et al., 2016; Richardson et al., 2019). Adjustments therefore form an important component of climate change assessment and necessitate the use of climate model integrations to simulate the radiative response of tropospheric and land surface changes to TOA energy imbalance, in addition to the traditional diagnostic of IRF or SARF,

which can be calculated using offline radiative transfer codes or simplified expressions (e.g. Hansen et al., 1988; Myhre et al., 1998; Etminan et al., 2016; Meinshausen et al., 2020). This makes ERF considerably more computationally-expensive to estimate and introduces more model diversity driven uncertainty. The use of different methods to calculate ERF further complicates inter-model comparison, with some studies opting to diagnose the forcing from fixed sea-surface temperature (SST) and sea ice simulations (Hansen et al., 2005), or alternatively, by regressing TOA irradiance against global surface

temperature change (Gregory et al., 2004; see Forster et al., 2016).

For $CO_2$, inter-model spread in ERF remains an ongoing issue. Smith et al. (2020a) report a 4x$CO_2$ ERF range of 7.3-8.9 W m$^{-2}$ for 17 CMIP6 (Coupled Model Intercomparison Project Phase 6; Eyring et al., 2016) models contributing to the Radiative Forcing Model Intercomparison Project (RFMIP; Pincus et al., 2016), which aims to achieve accurate characterisation of ERF

through consistent diagnosis with the fixed-SST method (Forster et al., 2016). Whilst this spread has been reduced compared to earlier analysis of 13 CMIP5 models (Kamae and Watanabe 2012; see Smith et al., 2020a Fig. 5), identifying and remedying the exact nature of $CO_2$ ERF diversity is an active area of research (e.g., Soden et al., 2018; Pincus et al., 2016; Smith et al., 2020a). Several studies show that model differences in the magnitude of IRF contributes significantly (e.g., Zhang and Huang 2014; Chung and Soden 2015b; Andrews et al., 2015), arising either from radiative transfer parameterisation error (e.g., Collins

et al., 2006; Pincus et al., 2015) and/or differences in model base-state climatology (Pincus et al., 2020; Jeevanjee et al., 2021). Recently, He et al., (2023) more specifically attribute this base-state dependence to stratospheric temperature. They report a

significant correlation between $4xCO_2$ IRF and 10 hPa air temperature in CMIP5/6 models, demonstrating that biases in stratospheric temperature play a leading role in causing inter-model $CO_2$ IRF spread. Given that IRF accounts for around 60% of $CO_2$ ERF and that stratospheric cooling is its dominant adjustment (Myhre et al., 2013; Smith et al., 2018), examining potential causes of model differences in stratospheric temperature presents a clear opportunity to further current understanding.

One such cause could relate to stratospheric $O_3$ – a key constituent in modulating stratospheric temperature. Depending on the treatment of stratospheric chemistry, models adopt a range of methods to generate $O_3$ fields using either an interactive chemistry scheme, a simplified online scheme or a prescribed pre-simulated dataset. Consequently, the resulting spatial structure and regional distribution of concentrations can differ substantially. Keeble et al. (2021) evaluate long-term $O_3$ trends in 22 CMIP6 models and find poor agreement in the simulation of pre-industrial total column ozone (TCO), with a variation from 275 to 340 DU between 60˚N-60˚S. Further, a ~20 DU range is observed between 10 of the models that prescribe stratospheric $O_3$ according to the CMIP6 $O_3$ dataset (Checa-Garcia, 2018), highlighting that even the model-specific implementation of common input can lead to significant differences in TCO.

Here, we perform idealised experiments (Section 2) to investigate the role of stratospheric $O_3$ as a driver of inter-model diversity in stratospheric temperature, and hence its role as a driver of spread in $CO_2$ ERF. First, we examine $4xCO_2$ ERF and compare our results to previous estimates, with a particular focus on the diagnosis of IRF and $T_{Strat}$ (Section 3). We then investigate the impact of stratospheric $O_3$ specification on each component of $4xCO_2$ ERF (Section 4).

## 2 Models, experiments and methods

We use atmosphere-only simulations from NorESM2-MM (Seland et al., 2020) to calculate ERF following an abrupt quadrupling of $CO_2$ relative to pre-industrial (1850) conditions (see Text S1 in the Supplement for further detail on model configuration). This model is used to perform a baseline (control) integration and a perturbed ($4xCO_2$) integration using prescribed SST and sea-ice extent climatologies; hence we use the fixed-SST method to diagnose forcing as recommended by RFMIP (Pincus et al., 2016) whereby ERF is calculated as the difference in TOA net radiative flux between the perturbed and control simulations. Integrations are run for 30 years, with years 6 to 30 used for analysis in Section 3. This simulation length was chosen to allow for better comparison of our results against the 30-year NorESM2-MM $4xCO_2$ ERF experiments of Smith et al. (2020a).

We perform two further $4xCO_2$ ERF experiments whereby stratospheric $O_3$ is increased by 50% (Strat $O_3$x1.5) and decreased by 50% (Strat $O_3$x0.5) relative to its pre-industrial concentration. Considering the substantial range in pre-industrial TCO noted by Keeble et al. (2021, Figure A3), we choose such a large, idealised increase/decrease in attempt to cover a broader range of stratospheric $O_3$ than shown by CMIP6 models, thus any effect on $4xCO_2$ ERF would likely be amplified in comparison.

Note that in each ERF experiment the $O_3$ increase (or decrease) is applied to both the control and $4xCO_2$ simulation so that the new $O_3$ field acts exclusively to alter the base-state atmosphere and does not act as a forcing itself. As in the 'standard' $4xCO_2$ ERF experiment described above, stratospheric $O_3$ fields are prescribed using output from the Community Earth System Model version 2 - Whole Atmosphere Community Climate Model version 6 (CESM2-WACCM6; Gettelman et al., 2019) as zonally-averaged 5 day fields (Fig. S1). A linearly varying tropopause (from 100hPa at the equator to 300 hPa at the poles) is used to delineate the stratosphere and troposphere (Soden et al., 2008, Smith et al. 2018). $O_3$ concentrations above this boundary are multiplied by 1.5 and 0.5 to increase and decrease levels by 50%, respectively. These simulations are run for 15 years to reduce computational expense, with years 6 to 15 of each integration used for analysis (Section 4). Table S1 summarises all experiments.

IRF is calculated using the Parallel Offline Radiative Transfer (PORT; Conley et al., 2013) code. This code isolates the radiative transfer scheme employed by NorESM2-MM (i.e., RRTMG, Iacono et al., 2008) to provide stand-alone offline radiation diagnostics. It is used here to perform two sets of radiative transfer calculations for each experiment listed in Table S1; a baseline (control) simulation and a perturbed ($4xCO_2$) simulation, which are both run using a year's worth of climatology from the corresponding ERF control integration (i.e., the first 12 months of its output). Simulations are then run for 12 months to diagnose annual-mean IRF as the difference in TOA net radiative flux between the perturbed and control run.

Corresponding radiative adjustments are quantified using radiative kernels (Soden et al., 2008). Summarising the more detailed description given by Smith et al. (2018), these characterise the change in TOA radiative flux $\Delta R$ (either shortwave or longwave) following a unit change in a state variable ($\Delta x$), e.g., temperature, surface albedo or water vapour. They are constructed by running a climate model's offline radiative transfer code twice, once with a baseline climatology and again with a unit change in $x$ to calculate $\Delta R$. The radiative kernel ($K_x$) is given by:

$$K_x = \frac{\partial R}{\partial x}. \tag{2}$$

The corresponding adjustment ($A_x$) is then quantified as:

$$A_x = K_x(x_p - x_c), \tag{3}$$

whereby $x_p - x_c$ represents the difference in $x$ between the perturbed and control atmosphere-only climate model integrations, respectively. Here, $A_x$ is calculated using output from NorESM2-MM with radiative kernels derived from three models: the Community Earth System Model 1-Community Atmosphere Model 5 (CESM-CAM5, Pendergrass et al., 2018), the Hadley Centre Global Environment Model 3-GA7.1 (HadGEM3-GA7.1, Smith et al., 2020b), and the European Centre for

Medium-Range Weather Forecasts (ECMWF)-Oslo model (Myhre et al., 2018). We use kernels to calculate all adjustments given in Eq. (1) except for $A_c$ which cannot be directly calculated from a kernel since the radiative effects of clouds are too nonlinear (Soden et al., 2008). To estimate $A_c$ we calculate the difference between all-sky and clear-sky TOA ERF (i.e., the change in cloud radiative effect; $\Delta$CRE) and then modify this to correct for cloud masking of the clear-sky $4\times CO_2$ IRF and adjustment response (see Soden et al., 2008 also Smith et al., 2018; Smith et al., 2020a) This differs from alternate approaches

used to calculate $A_c$ such as the approximate partial radiative perturbation (APRP) method, which estimates shortwave cloud responses from climate model diagnostics (Zelinka et al., 2014; Smith et al., 2018) and the offline partial radiative perturbation (PRP) method, where cloud radiative effects are estimated using an offline radiative transfer code and model cloud fields (e.g. Smith et al., 2020a). Additionally, we also use kernels to calculate the adjustment due to surface temperature change ($A_{T_S}$) as in Smith et al. (2018), since land surface temperatures are allowed to respond to the forcing in our simulations given the

difficulty in prescribing fixed surface temperatures (Forster et al., 2016). Several studies also follow this approach whereby the calculation of ERF includes the radiative response of land surface warming or cooling (e.g., Hansen et al., 2005; Forster et al., 2016; Smith et al., 2018; Smith et al., 2020a). Generally, methods that correct for this produce a slightly larger ERF following a $CO_2$ perturbation (Smith et al., 2020a; Andrews et al., 2021). Subtracting $A_{T_S}$ from the ERF could provide a land surface warming corrected forcing (following Smith et al., 2020a), however we do not calculate this here. Instead, we report

the magnitude of $A_{T_S}$ to inspect any change in its value with each $O_3$ experiment. Further, we use the same tropopause definition as above to delineate $A_{T_{Strat}}$ and $A_{T_{Trop}}$.

## 3 The importance of a direct calculation of IRF and the dependence of $A_{T_{Strat}}$ on radiative kernel choice

Figure 1a (purple bars) shows the resulting NorESM2-MM ERF, IRF and adjustments. For comparison, corresponding data from the NorESM2-MM $4\times CO_2$ ERF experiment of Smith et al. (2020a) is also shown (green bars). As expected, the magnitude

of ERF is near-equal in each experiment, at 8.40 W m$^{-2}$ (purple bar) and 8.38 W m$^{-2}$ (green bar). The difference of 0.02 W m$^{-2}$ is likely attributable to differences in the time-period used to average model output, or to the use of alternate initial conditions and computing machine architecture given that all other aspects of simulation design were implemented identically (see Section 2 and Smith et al., 2020a, Section 2).


a)

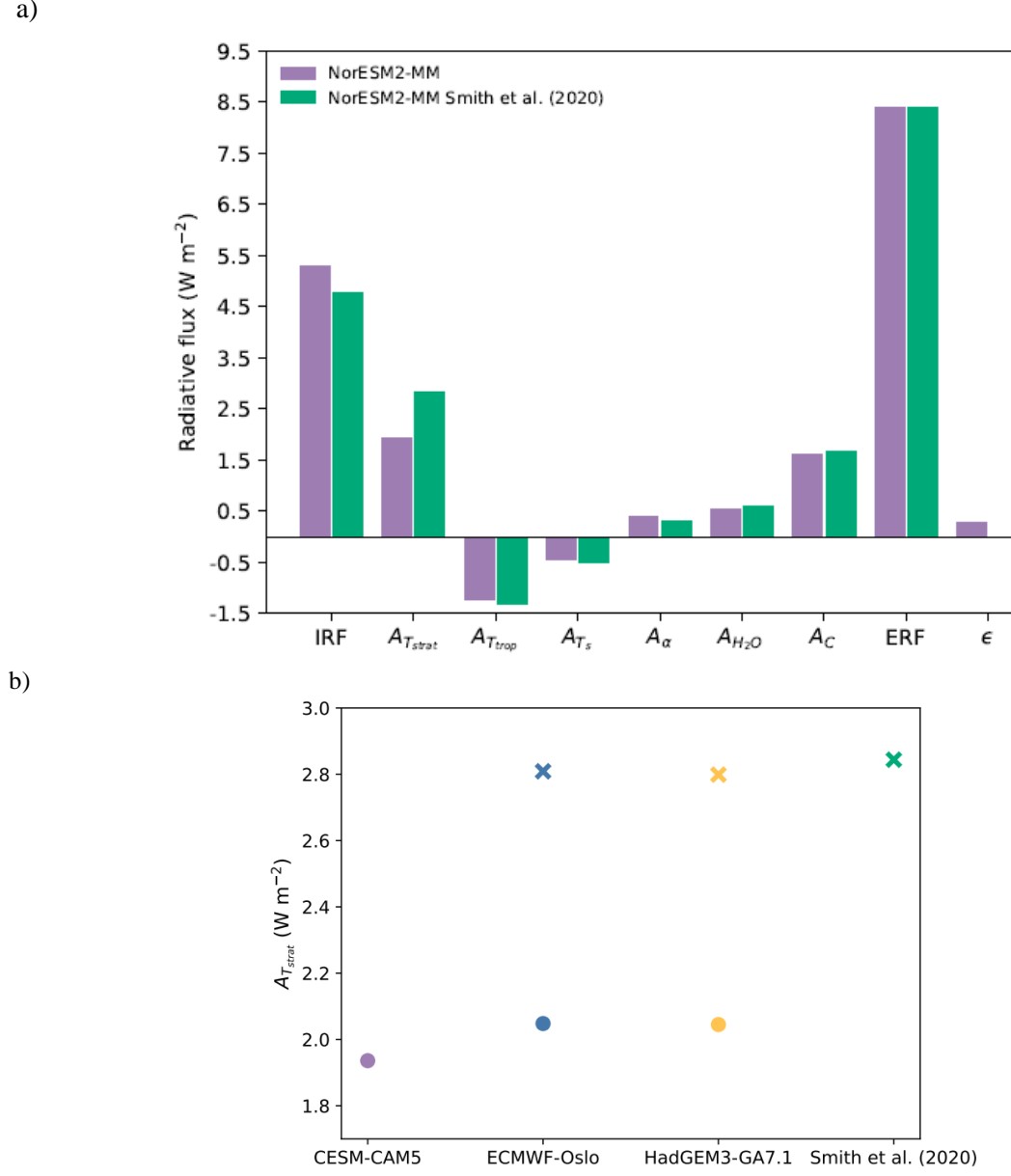

b)

Figure 1: a) NorESM2-MM 4xCO$_2$ ERF, IRF, adjustments and residual term $\epsilon$ (purple bars), with IRF calculated using PORT and adjustments diagnosed using the CESM-CAM5 kernels. Green bars show corresponding data from Smith et al. (2020a), whereby the ERF for the same perturbation has been calculated from 30-year simulations of NorESM2-MM with non-cloud adjustments calculated using the HadGEM3-GA7.1 radiative kernels and SW and LW cloud adjustments calculated using the APRP and PRP method, respectively with IRF estimated as the residual of ERF minus total

adjustments, hence there is no specific $\epsilon$ term as this is aliased into the IRF. b) Comparison of NorESM2-MM $A_{T_{Strat}}$ when calculated using different radiative kernels. Filled circles represent $A_{T_{Strat}}$ calculated following the methodology outlined in Section 2, whereby NorESM2-MM output is interpolated onto the given radiative kernel pressure levels but not extrapolated to kernel pressure levels outside of the NorESM2-MM uppermost and lowermost pressures. Crosses represent the magnitude of $A_{T_{Strat}}$ if such extrapolation is performed using the ECMWF-Oslo (blue cross) and HadGEM3-GA7.1 (yellow cross) kernels along with the value given by Smith et al. (2020a) (green cross, which uses the HadGEM3-GA7.1 kernel).


Figure 1a further shows that the magnitude of IRF varies notably, demonstrating a dependence on the diagnostic method of choice. When calculated directly using PORT, the IRF is 0.54 W m$^{-2}$ larger than when estimated as the difference between ERF and the sum of adjustments (as in Smith et al. 2020a), comparing 5.30 W m$^{-2}$ (purple bar) and 4.76 W m$^{-2}$ (green bar).
This demonstrates the necessity of using a model's own radiative transfer code to calculate IRF and highlights the possibility for error in studies that derive this forcing as a residual. Directly calculating the IRF also permits the calculation of $\epsilon$ (see Eq.1) to analyse the magnitude of non-linearities that are not accounted for by the kernel-derived adjustments. Our $\epsilon$ value (calculated as ERF-IRF-$\Sigma A_x$) is 3% (0.27 W m$^{-2}$) of the ERF (8.40 W m$^{-2}$) and therefore well within the 10% guideline given by Shell et al., (2008). However, when spectrally split, the shortwave $\epsilon$ is 33% (-0.47 W m$^{-2}$) of the shortwave ERF (1.42 W
m$^{-2}$) and works to partially counteract a longwave $\epsilon$ of 0.74 W m$^{-2}$ (which itself is 11% of the longwave ERF of 6.98 W m$^{-2}$). This finding also extends to our analysis of the clear-sky 4xCO$_2$ forcing and adjustments (not shown). Such larger residuals can possibly be explained by the collapse of linear behaviour for a large perturbation like 4xCO$_2$ (see Jonko et al., 2012, Smith et al., 2020b). We further note the close agreement in $A_c$, which occurs despite the use of different methods to calculate it. In Smith et al. (2020a) shortwave and longwave $A_c$ are estimated separately using the APRP approach and offline monthly-mean
PRP calculations, respectively. In our approach, $A_c$ is calculated using the adjusted CRE method (see Section 2).

The stratospheric temperature adjustment is strong and positive as anticipated due to the process of stratospheric cooling following an increase in CO$_2$ concentration (e.g. Myhre et al., 2013; Smith et al., 2018; Forster et al., 2021). However, there is a clear difference when comparing the value reported here (1.94 W m$^{-2}$, purple bar) against Smith et al. (2020a; 2.84 W m$^{-2}$, green bar). Because $A_{T_{Trop}}$ is similar between both experiments (-1.23 W m$^{-2}$ vs -1.32 W m$^{-2}$) it can be deduced that the
difference in magnitude of $A_{T_{Strat}}$ is not predominantly driven by the choice of tropopause definition (in Smith et al. (2020a) this is based on the World Meteorological Organization definition of a lapse-rate tropopause). Instead, the difference in $A_{T_{Strat}}$ stems from the use of different radiative kernels (i.e., CESM-CAM5 vs HadGEM3-GA7.1) and the method of applying model output in the $A_{T_{Strat}}$ calculation. For our derivation of $A_{T_{Strat}}$ we interpolate NorESM2-MM output to the 30 CESM-
CAM5 kernel pressure levels, where 3.64 hPa is the highest level. Even though this is a 'low-top' kernel, this matches the

highest level of NorESM2-MM output meaning that the use of this kernel in the $A_{T_{Strat}}$ calculation captures all of the stratospheric cooling occurring in NorESM2-MM following a 4xCO$_2$ perturbation. Alternatively, Smith et al. (2020a) use the HadGEM3-GA7.1 radiative kernel, which itself has been interpolated from a native vertical resolution of 85 pressure levels (up to around 0.005 hPa) to the standard 19 CMIP6 pressure levels, with an upper bound of 1 hPa. Smith et al. (2020a) derive

$A_{T_{Strat}}$ by using model output that has been interpolated and extrapolated to the 19 CMIP6 pressure levels. This therefore extends stratospheric temperatures in NorESM2-MM above the model's highest level of 3.64 hPa to 1 hPa. Whilst this method better accounts for outgoing radiation emitted to space from the upper stratosphere for each unit change in temperature, it does not represent the actual adjustment modelled by NorESM2-MM.

Figure 1b (filled circles) further demonstrates this issue by comparing the magnitude of $A_{T_{Strat}}$ calculated by applying our NorESM2-MM output to two additional kernels: ECMWF-Oslo and HadGEM3-GA7.1. The ECMWF-Oslo kernel has 60 pressure levels, with a high resolution in the stratosphere extending to 0.1 hPa, and as described above, the HadGEM3-GA7.1 kernel utilises the standard CMIP6 19 pressure levels. When we interpolate (but do not extrapolate) NorESM2-MM output onto these pressure levels, the use of both ECMWF-Oslo and HadGEM3-GA7.1 results in an adjustment similar to that given

by CESM-CAM5, at 2.05 W m$^{-2}$ (blue and yellow filled circles). However, when NorESM2-MM output is both interpolated and extrapolated to the upper stratospheric levels of the ECMWF-Oslo and HadGEM3-GA7.1 kernels, the adjustment is notably stronger (and in closer agreement with Smith et al. 2020a) at around 2.80 W m$^{-2}$ (blue and yellow crosses). The importance of the vertical resolution of stratosphere has been stated previously in studies quantifying the magnitude of $A_{T_{Strat}}$ to a CO$_2$ forcing. Notably, Smith et al. (2018) demonstrate that disagreement in 2xCO$_2$ $A_{T_{Strat}}$ is dependent on whether

a given kernel has high stratospheric resolution (e.g., ECMWF-Oslo) and if the model output is also highly resolved in the stratosphere. Smith et al. (2020b) further report that kernels based on a high-top atmospheric model with a large number of native pressure levels have a pronounced increase in the magnitude and rate of emitted radiation at 5 hPa and 1 hPa. Here, the difference between the 'extrapolated' (blue and yellow crosses) and 'not-extrapolated' $A_{T_{Strat}}$ values (blue and yellow circles) in Fig. 1b infers that around 0.75 W m$^{-2}$ of 'additional' stratospheric temperature adjustment occurs between the model top and

the upper pressure limit of the ECMWF-Oslo and HadGEM3-GA7.1 kernels (0.1 hPa and 1 hPa, respectively). This therefore supports previous studies that highlight the significance of vertical stratospheric resolution on $A_{T_{Strat}}$, and further demonstrates that the choice and method of applying a radiative kernel can substantially impact results. Opting to use a radiative kernel that has been constructed from the same atmospheric model as the CO$_2$ forcing simulations in question will more accurately represent the magnitude of $A_{T_{Strat}}$ simulated within that given model. This also ensures that the calculation of $A_{T_{Strat}}$ is based

entirely on one underlying radiative transfer code, which eliminates any uncertainty in the magnitude of $A_{T_{Strat}}$ that could occur if the kernel and model output were derived from two different parameterisations. If a radiative kernel is not available for a given model, or a kernel needs to be applied across multiple models to evaluate inter-model spread, then it could be more

suitable to not extrapolate data outside of each model's native vertical bounds. However, the best use of kernels is likely quite case specific.

## 4 Stratospheric $O_3$ experiments

### 4.1 Impact on stratospheric temperature

$O_3$ plays an important role in driving the thermal structure of the stratosphere due to strong absorption of ultraviolet radiation and absorption and emission of thermal-infrared (TIR) radiation. Figure 2 (left) shows the effect of a 50% increase in stratospheric $O_3$ concentration on zonal-mean atmospheric temperature in the control integration of NorESM2-MM. A strong increase in stratospheric temperature is evident, consistent with enhanced absorption of solar radiation and hence enhanced solar heating rates. The peak increase in temperature occurs in the lower stratosphere centred across the equatorial region, co-located with high insolation. Here, the maximum $\Delta T$ reaches 5.8 K. Similarly, decreasing stratospheric $O_3$ concentration by 50% results in reduced absorption of solar radiation, reduced solar heating rates and a strong cooling of the stratosphere (Fig. 2, right). As above, the peak temperature decrease occurs in the lower stratosphere across the equator, with a maximum $\Delta T$ of -9 K. The impact of reduced stratospheric $O_3$ also propagates into the troposphere (primarily between 70°-90°N/S), due to more downward solar irradiance reaching the lower levels of the atmosphere where enhanced absorption and heating can take place. Considering the high correlation between 4x$CO_2$ IRF and 10 hPa air temperature reported by He et al. (2023), we note that at this level in particular $\Delta T$ largely increases by $\geq$ 3 K in the 'Strat $O_3$x1.5' case (Fig 2., left) and largely decreases by $\geq$ 4 K in the 'Strat $O_3$x0.5' case (Fig 2., right).

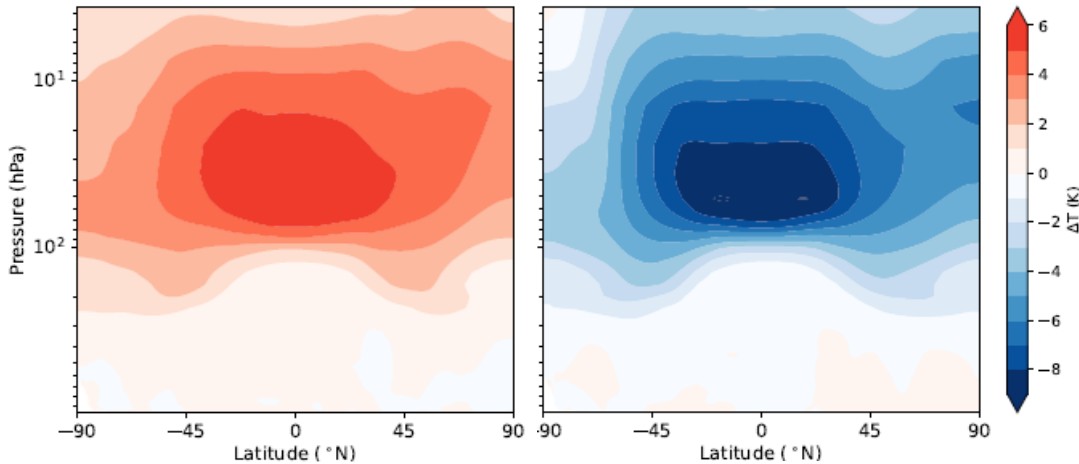

Figure 2: Zonal-mean difference in atmospheric temperature between the control integration of 'Strat $O_3$x1.5' and the control integration of the 'standard' 4xCO$_2$ ERF simulation (left) and between the control integration of 'Strat $O_3$x0.5' and the control integration of the 'standard' 4xCO$_2$ ERF simulation (right). Note that for display purposes model output on hybrid-sigma levels has been interpolated onto global-mean NorESM2-MM pressure levels.

## 235   4.2 Impact on 4xCO$_2$ ERF and components

Figure 3 compares ERF, IRF and the individual adjustments for the 'standard' 4xCO$_2$, 'Strat $O_3$x1.5' and 'Strat $O_3$x0.5' experiments. As shown, increasing stratospheric $O_3$ by 50% has negligible impact on the magnitude of 4xCO$_2$ ERF in NorESM2-MM, resulting in a near identical forcing (of 8.47 W m$^{-2}$) compared to the 'standard' case (dark-orange bar). Similarly, the effect of decreasing stratospheric $O_3$ by 50% has a marginal effect on 4xCO$_2$ ERF, increasing the forcing by just

0.23 W m$^{-2}$ relative to the 'standard' case. Evidently, the impact of increased/decreased $O_3$ concentration on stratospheric temperature (Fig. 2) does not result in a marked effect on ERF. In all three cases NorESM2-MM simulates a considerably larger ERF than the 17 CMIP6 multi-model mean of 7.98 W m$^{-2}$ reported by Smith et al. (2020a).


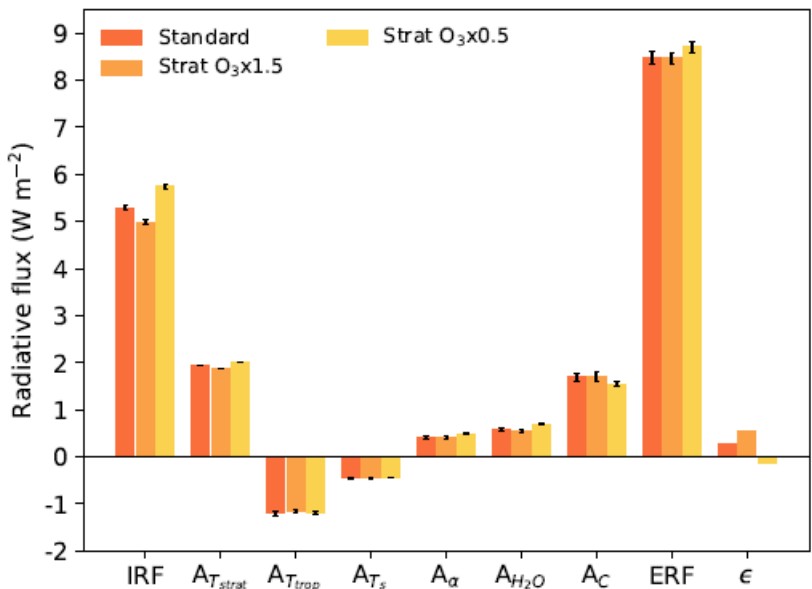

Figure 3: Comparison of NorESM2-MM 'standard' 4xCO$_2$ ERF, IRF, adjustments and residual term against the ERF, IRF, adjustments and residual term diagnosed from the 'Strat O$_3$x1.5'and 'Strat O$_3$x0.5' experiments. All adjustments are derived using the CESM-CAM5 kernels and all components are calculated from the average of years 6-15 of NorESM2-MM output, hence the values for the 'standard' 4xCO$_2$ case shown here differ slightly to those shown in Fig. 1. Error bars show the standard error of the mean of each component, note that we do not compute an error estimate for $\epsilon$

Analysis of $A_{T_{Strat}}$ further demonstrates that these experiments do not cause a significant effect on the magnitude of temperature adjustment throughout the stratosphere, producing just a 3% decrease and 4% increase in this component for 'Strat O$_3$x1.5' and 'Strat O$_3$x0.5', respectively. This corroborates experiments from He et al., (2023; Figure S6) that compare the size of stratospheric temperature adjustment after a quadrupling of CO$_2$ from two different base-states; the multi-model ensemble-mean difference in their adjustment is just -0.03 W m$^{-2}$ (although the model range is considerably larger at around

0.5 W m$^{-2}$). As described by Shine and Myhre (2020), the magnitude of stratospheric temperature adjustment is driven by the balance between enhanced stratospheric emittance at the TOA and enhanced stratospheric absorptance of upwelling irradiance from the troposphere, which lead to a cooling and warming of the stratosphere, respectively. Following an increase in CO$_2$, enhanced TOA emission is greater than enhanced stratospheric absorptance because upwelling irradiance largely emanates from the cold upper troposphere with a low effective emitting temperature (e.g., Ramaswamy et al., 2001). Subsequently, the

stratosphere cools, leading to a decrease in longwave emission to space, which in turn, strengthens the TOA forcing. Given that the magnitude of $A_{T_{Strat}}$ shows little variation in Figure 3, we can infer that the net effect of increased emissivity versus increased absorptance is similar across each experiment, leading ultimately, to a cooler stratosphere and a stronger TOA

radiative imbalance in all cases. Whilst increased/decreased stratospheric $O_3$ effects the degree of spectral overlap with $CO_2$ (discussed further below) and the vertical profile of stratospheric absorption and emission, it is apparent that the base-state of the stratosphere is not a significant factor in determining the magnitude of our $4xCO_2$ temperature adjustments.

Evidently, the largest impact occurs on IRF which increases by 8% and decreases by 6% when stratospheric $O_3$ is reduced and enhanced, respectively, relative to the 'standard' experiment. The IRF across all three experiments ranges from $4.98 - 5.74$ W m$^{-2}$ resulting in a spread of around 0.8 W m$^{-2}$. This is smaller than the spread of 2 W m$^{-2}$ (ranging from around $5 - 7$ W m$^{-2}$) reported by He et al. (2023) for offline 'double-call' experiments of $4xCO_2$ IRF calculated with a single radiative transfer code and base-states from the Atmospheric Model Intercomparison Project (AMIP) for 12 CMIP5/6 models (hence their spread is due only to differences in base-state). However, we find closer agreement between our IRF sensitivity to 10 hPa base-state temperature and He et al. (2023), who found a near -0.1 W m$^{-2}$ K$^{-1}$ relation between the spread in offline double-call experiments and air temperature at this level (Figure 1c in He et al., 2023). From Figure 2 it can be inferred that this matches very well with the $\sim\geq 7$ K difference in base-state 10 hPa temperature between 'StratO$_3$x1.5' and 'StratO$_3$x0.5' and the corresponding 0.8 W m$^{-2}$ spread in IRF.

The effect on IRF can be explained principally by the impact of $O_3$ increases/decreases on base-state stratospheric temperature and secondarily by the spectral overlap of $CO_2$ and $O_3$[1]. In the 'Strat $O_3$x0.5' case for example, the reduced $O_3$ concentration induces a colder base-state stratosphere, which reduces the emission of outgoing TIR irradiance at the TOA and makes the radiative impact of a $4xCO_2$ perturbation more potent. The reverse is true in the 'Strat $O_3$x1.5' case. In relation to spectral overlap, $O_3$ itself possesses two fundamental absorption bands in the TIR around 9.6 µm and 14.27 µm, with a relatively strong band formed by overtone and combination transitions centred at 4.75 µm. As stratospheric $O_3$ concentrations increase, TIR absorption at these wavelengths also increases to an extent that depends on the level of band saturation and the abundance of other gases absorbing at these wavelengths. The opposite occurs if stratospheric $O_3$ concentrations decrease. For $CO_2$, the main TIR bands lie in the window regions of the $H_2O$ spectrum, with absorption centred at 4.3 µm and 15 µm (the latter of which is highly significant due to its proximity to the peak of blackbody distribution for the Earth's effective emitting temperature). Weaker bands also occur near 10 µm. Regions of spectral overlap between $O_3$ and $CO_2$ therefore arise at several wavelengths; at 15 µm the strength of $CO_2$ absorption largely masks the radiative effect of $O_3$ at 14.27 µm and absorption by both gases at 4.75 µm and 4.3 µm has little impact given their location further away from the peak of Earth's blackbody distribution. However, decreased stratospheric $O_3$ concentration leads to weakened absorption at 9.6 µm that allows for enhanced absorption by $CO_2$ at 10 µm. Likewise, increasing stratospheric $O_3$ concentration results in strengthened 9.6 µm absorption that mutes $CO_2$. Combining the effect of base-state stratospheric temperature and spectral overlap, a $4xCO_2$ perturbation

---

[1]Tests performed by the GENLN2 line-by-line (Myhre et al., 2006) show that a decrease in temperature of 2 K across the whole stratosphere leads to a 0.16 W m$^{-2}$ increase in $4xCO_2$ IRF, whilst a 50% reduction in stratospheric $O_3$ leads to a 0.07 W m$^{-2}$ increase in $4xCO_2$ IRF.

therefore results in an enhancement of IRF in the 'Strat $O_3$x0.5' case relative to the 'standard' experiment. Correspondingly, an increase in stratospheric $O_3$ has the opposite (albeit evidently weaker) effect.

As successive adjustments to the 4x$CO_2$ perturbation either strengthen or weaken IRF, the radiative impact of both $O_3$ experiments is either enhanced or reduced according to the sign and magnitude of each $A_x$ term in Figure 3 (see also Table S2). Although somewhat minorly, $A_{T_{Strat}}$, $A_{T_S}$, $A_\alpha$, $A_{H_2O}$ (and $A_{T_{Trop}}$ for 'Strat $O_3$x0.5') work to strengthen the difference of each experiment relative to the 'standard' case (i.e., enhancing the TOA radiative impact for 'Strat $O_3$x0.5' and decreasing the TOA radiative impact for 'Strat $O_3$x1.5'). Conversely, $A_c$ (and $A_{T_{Trop}}$ for 'Strat $O_3$x1.5') have the opposite effect; for these terms the TOA radiative impact for 'Strat $O_3$x0.5' is reduced relative to the 'standard' case and enhanced for 'Strat $O_3$x1.5'.

A closer inspection of $A_c$ (Fig. S2a) reveals that a significant part of this offsetting behaviour stems from $\Delta$CRE. This is more positive for 'Strat $O_3$x1.5' compared to the 'standard' and negative for 'Strat $O_3$x0.5', implying for the latter that the presence of clouds reduces the 4x$CO_2$ ERF. Interestingly, this phenomenon occurs due to a decidedly stronger clear-sky SW ERF (Table S3), which reduces the difference between all-sky and clear-sky forcing (Fig. S2b) resulting in a net negative $\Delta$CRE when combined with the LW (Fig. S2c, S2a). In general, in all cases the 4x$CO_2$ perturbation induces a similar zonal-mean decrease in low cloud fraction/increase in high cloud fraction (Fig S3a). However, relative to the 'standard' base-state, both $O_3$ experiments clearly have the greatest impact on cloud fraction in the upper troposphere at almost all latitudes, whereby coverage decreases in 'Strat $O_3$x1.5' and increases in 'Strat $O_3$x0.5' (Fig. S3b). As expected for high clouds, this change appears to have the strongest influence on thermal fluxes, resulting in a weaker and stronger masking of LW IRF for 'Strat $O_3$x1.5' and 'Strat $O_3$x0.5', respectively (Table S3). However, for the net IRF and adjustment terms (Fig. S2a) the effect of cloud masking shows little variation between each experiment and therefore plays a less significant role in driving the differences observed in $A_c$.

Finally, we note that the residual term, $\epsilon$, also exhibits offsetting behaviour and works to counterbalance the initial radiative impact of each $O_3$ experiment the most (Fig. 3, Table S2), implying that non-linearity in Eq.1 is largely responsible for dampening the differences with regard to IRF. Overall, in each experiment summation of IRF, $A_x$ and $\epsilon$ results in strikingly similar ERFs for the 'standard' and 'Strat $O_3$x1.5' cases, and a slightly larger ERF for 'Strat $O_3$x0.5'.

In discussion of the potential climate implications of their findings, He et al. (2023) suggest that $O_3$ depletion since the 1970s could have led to a strengthening of TOA $CO_2$ IRF due to the cooling of the lower stratosphere associated with $O_3$ loss. They theorise that the combined effect of $O_3$ depletion and $CO_2$ increase should produce a larger $CO_2$ ERF and a greater surface warming than model experiments that impose these perturbations separately. They calculate the indirect surface warming effect of $O_3$ loss by differencing surface temperature anomalies between two such sets of experiments (historical forcing between

1985-2014 vs the sum of all historical forcings between 1985-2014 imposed independently) and infer that the sign and spatial distribution of the nonlinear warming contribution of $O_3$ loss to $CO_2$ IRF is consistent with the base-state dependence of IRF. As shown above, we demonstrate that a highly idealised reduction in stratospheric $O_3$ does lead to an enhancement of $4xCO_2$
330    IRF. However, we find that this does not significantly affect the magnitude of ERF.

## 5 Conclusions

Here we demonstrate that accurate calculation of IRF requires the use of host-model radiative transfer calculations, which can be computed either offline or online using double-call simulations. As noted elsewhere (e.g., Chung and Soden 2015b), we
335    encourage modelling centres to make this diagnostic available with their simulations. Inferring IRF indirectly as the residual of ERF and the sum of adjustments can result in the erroneous estimation of its magnitude, which introduces further uncertainty into the exact nature of inter-model spread in $CO_2$ ERF. We further show that increasing and decreasing stratospheric $O_3$ by 50% results in a strong warming and cooling of stratosphere, with the peak change in temperature in each experiment reaching around 6 K and -9 K, respectively. Despite the sizeable effect on stratospheric temperature, these highly idealised changes in
340    $O_3$ concentration do not result in a correspondingly large spread in the magnitude of stratospheric temperature adjustment or $4xCO_2$ ERF. Instead, these experiments demonstrate a dominant impact on the magnitude of IRF, primarily due to the impact on base-state stratospheric temperature with an ancillary effect from spectral overlap of $CO_2$ and $O_3$. Given that such large changes in stratospheric $O_3$ do not yield a significant impact on $4xCO_2$ ERF, our results suggests that inter-model differences in stratospheric $O_3$ concentration are not predominantly responsible for inter-model spread in $CO_2$ forcing.

345

*Code availability.* NorESM2 (tag release-noresm2.0.6) can be downloaded from https://github.com/NorESMhub/NorESM (Seland et al., 2020).

*Data availability:* The CESM-CAM5 radiative kernels are freely available at https://zenodo.org/records/997902 (Pendergrass et al., 2018). The ECMWF-Oslo kernels are freely available at https://github.com/ciceroOslo/Radiative-kernels.git (Myhre et al., 2018). The HadGEM3-GA7.1 kernels are freely available at https://doi.org/10.5281/zenodo.3594673 (Smith et al., 2020b).

*Author contributions.* GM and REB designed the study. REB performed the simulations, with supporting calculations performed by GM and ØH. REB and GM analysed the data. REB produced the figures and was the primary writer of the manuscript with contributions from GM, DO, ØH and MS.

*Competing interests.* At least one of the (co-)authors is a member of the editorial board of Atmospheric Chemistry and Physics.

*Acknowledgements.* We thank Christopher J. Smith for his input on technical aspects of this manuscript and for his review and comments. We also thank Marit Sandstad for her technical advice on model simulations. REB and GM were supported by the European Union's Horizon 2020 research and innovation programme under grant agreement No 820829 (CONSTRAIN).

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
