# Peer review of "Investigating the role of stratospheric ozone as a driver of intermodel spread in CO2 effective radiative forcing."

_EGUsphere, 2024_

## Author Comment (AC1)

We thank this reviewer for their positive and constructive comments and their helpful advice for improving the manuscript. Our responses are given in the blue font. Where necessary, we refer to the revised version of the manuscript.

Review of egusphere-2024-111, entitled "Investigating the role of stratospheric ozone as a driver of inter-model spread in $CO_2$ effective radiative forcing" by Byrom et al.

**General comments**

The submitted manuscript investigates stratospheric ozone as a cause of inter-model differences in stratospheric temperature, and hence its role as a contributing factor to inter-model spread in $CO_2$ radiative forcing. The work aims to explore whether the stratospheric temperature dependence of instantaneous radiative forcing (IRF) extends to effective radiative forcing (ERF). Using the Norwegian Earth System Model 2, the authors explore the impact of stratospheric ozone perturbation and further stratospheric temperature on the magnitude of 4xCO2 ERF and its components (IRF and rapid adjustments) via a series of well-designed fixed-SST simulations. The authors found that the systematic stratospheric ozone perturbations barely influence the spread of 4xCO2 ERF, although the effects of systematic stratospheric ozone perturbations clearly show in both the IRF and stratospheric adjusted radiative forcing (SARF) via their base-state dependence. Meanwhile, the authors also found considerable uncertainty in the IRF calculations with different methods, stemming from differences in both radiative transfer codes and the vertical coordinates (e.g., model top pressure and vertical resolutions). These results may help to better understand the large spread in the IRF and further ERF, hopefully reducing uncertainty in estimates of climate sensitivity. Overall, I found the manuscript well-organized, well-illustrated, and a good addition to the understanding of state-dependent radiative forcing. I would recommend Atmospheric Chemistry and Physics consider this article for publication after minor revisions.

**Major concern:**

The authors found that the systematic stratospheric ozone perturbations barely influence the spread of 4xCO2 ERF, although the effects of systematic stratospheric ozone perturbations clearly show in both the IRF and SARF via their base-state dependence. Based on Figure 3, it is clear that the cloud adjustment offsets the stratospheric temperature dependence of the IRF and SARF. I would suspect that most cloud adjustments come from cloud changes at higher altitudes. It would be good if the author could decompose cloud adjustment into contributions from high, mixed, and low clouds, probably following Soden and Vecchi (2011). It would be even better if the authors had ISCCP simulator results available (it is totally ok if not). If the cloud adjustment comes from high altitudes, as suspected, the author could probably get the stratospheric temperature-dependent ERF by simulations only perturbing ozone within the upper stratosphere in the future.

Soden, B. J., and G. A. Vecchi (2011), The vertical distribution of cloud feedback in coupled ocean-atmosphere models, Geophys. Res. Lett., 38, L12704, doi:10.1029/2011GL047632.

This is a good point, and we agree with the reviewer that more attention should be given to the apparent impact of the ozone increases/decreases on the cloud adjustment. Unfortunately, ISCCP simulator results are not available for these NorESM2 simulations, so we couldn't diagnose and decompose the cloud adjustment via that method.

However, following this comment (and similar comments made by the other reviewers) we decided to change our method for calculating the cloud adjustment to the adjusted cloud radiative effect method of Soden et al. (2008; https://journals.ametsoc.org/view/journals/clim/21/14/2007jcli2110.1.xml, also

see our response to Reviewer 1). As shown in the updated version of Figure 3 (see revised manuscript), the magnitude of the cloud adjustment in each experiment is now much more similar, demonstrating that this adjustment doesn't offset the impact of ozone increase/decreases on IRF and SARF as implied previously.

**Minor comments:**

Lines 15-16: Are the host-model radiative transfer calculations referring to online or offline double-call calculations with the same radiative transfer codes? If yes, the authors could probably encourage model centers to provide online double-call for their simulations.

We perform offline host-model radiative transfer IRF calculations and demonstrate the need for others to also use the host-model radiative transfer code to calculate IRF, but this could be achieved either online or offline. We have now added more text to the conclusion (lines 303-305) to make it clear that these calculations could be performed offline or online and have included a sentence to encourage modelling centres to provide IRF results:

*"Here we demonstrate that accurate calculation of IRF requires the use of host-model radiative transfer calculations, which can be computed either offline or online using double-call simulations. As noted elsewhere (e.g., Chung and Soden 2015b), we encourage modelling centres to make this diagnostic available with their simulations."*

Lines 16 and 81-82: I wondered why the authors chose to use the 50% increase and decrease in stratospheric $O_3$ concentration instead of doubling and halving stratospheric $O_3$ concentration.

This choice to go with a 50% increase and decrease was a relatively arbitrary choice, however we went with these perturbations in order to generate a somewhat similar effect on temperature either way (i.e., with an increase and with a decrease). A doubling and halving of $O_3$ concentration would also be interesting to look at.

Lines 17-19, 240-241 & 265-268: Is the effect of the spectral overlap of $CO_2$ and $O_3$ comparable to the effect of stratospheric temperature dependence? It would probably be interesting to have a simple test with LBL codes in the future.

We previously performed LBL tests (using the Oslo LBL code) to compare the relative importance of these effects. We found that a 50% reduction in $O_3$ leads to a +0.07 W m$^{-2}$ increase in 4x$CO_2$ IRF (from 4.2 W m$^{-2}$), whilst a reduction in the temperature of -2K across the whole stratosphere leads to a 0.16 Wm$^{-2}$ increase. Based on these results the effect of stratospheric temperature dependence is stronger than the effect of spectral overlap. Furthermore, as shown in Figure 2, decreasing stratospheric ozone by 50% results in widespread cooling of the stratosphere with ΔT values largely more negative than -3 K and peaking at -9 K. We have now added these results as a footnote on page 12 of the revised manuscript:

*"Tests performed by the GENLN2 line-by-line (Myhre et al., 2006) show that a decrease in temperature of 2 K across the whole stratosphere leads to a 0.16 W m$^{-2}$ increase in 4x$CO_2$ IRF, whilst a 50% reduction in stratospheric $O_3$ leads to a 0.07 W m$^{-2}$ increase in 4x$CO_2$ IRF."*

Lines 92-95: Can the authors provide more details for the PORT offline calculations? Are the calculations described here just offline double-call calculations? So, just two offline calculations using 1x and 4x CO2 concentrations with identical base-state from control simulation. Can the authors help to explain what "the simulations are run for 16 months with the last 12 months used to …" means?

Yes, this is correct – the PORT IRF calculation consists of two offline calculations using pre-industrial $CO_2$ concentrations and $4xCO_2$ concentrations that are each run with an identical atmospheric base state from the control simulation that is taken from the first year of output from the NorESM2 simulation.

The part that says, "the simulations are run for 16 months with the last 12 months used to …" was based on the recommendation in Conley et al. (2013): *"Running the model for 4 months prior to the 12-month period of the average is recommended, so that the stratospheric temperatures are in steady state before the time for a full-year (1 January–31 December) computation begins."*

We originally also used PORT to perform stratospheric temperature adjusted RF (fixed dynamical heating) calculations and followed the above advice to run the model for 16 months (from 1st January year 1 – 30th April year 2) and use the last 12 months of the output to calculate the stratospheric temperature adjusted calculation (i.e. from 1st May – 30th April). We also derived IRF using 16 month long simulations and took the last 12 months of the 16 month output to calculate IRF, however we realise that this is not necessary given that an instantaneous forcing should ideally be calculated immediately following the perturbation. We have now derived all IRF results using the first 12 months of the simulation (i.e., 1st January – 31st December) and have amended the methodology. Note however, that this did not impact the IRF result for the 'Standard' experiment and only marginally changed the net IRF for the 'O3x1.5' and 'O3x0.5' experiments (by 0.01 W m$^{-2}$).

Line 99: It would probably be better to use "temperature, water vapor, or surface albedo" instead of "stratospheric temperature, surface albedo or clouds", since there is no cloud kernel in the radiative kernels of Soden et al. (2008).

This has now been changed to read as "temperature, surface albedo or water vapour" (line 113).

Lines 141-144 & 157-159: It is great to see the conclusion that the magnitude of IRF varies notably between both experiments, demonstrating a dependence on the diagnostic method of choice, although this may not be very new. Even with the identical radiative kernel method, the IRF obtained from 17- and 19-levels are noticeably different. In particular, stratosphere adjustment has a strong dependence on the model top and probably vertical resolution. This is because the radiative flux perturbation due to the same temperature perturbation at higher altitudes (e.g., 1 hPa) within the upper stratosphere is larger than that of lower altitudes (e.g., 10 hPa) within the upper stratosphere. Meanwhile, the temperature cooling at higher altitudes within the upper stratosphere is usually also stronger than at lower altitudes within the upper stratosphere. Therefore, stratospheric adjustment (IRF) from 17-level kernels is smaller (larger) than that of 19-level kernels.

This is a good point to highlight that the magnitude of the IRF consequently varies between the use of the 17 and 19 level kernels.

Lines 144-145, 171-175 & 187-191: Since the accuracy of IRF calculation from the radiative kernel method depends on the differences of radiative transfer codes and the vertical coordinate resolution (ignoring the base-state dependence), here it would be great to isolate the contribution from the difference of radiative transfer codes by interpolating (and extrapolating if necessary for surface and probably for CAM5 kernels) the three radiative kernels onto the output resolution of NorESM2-MM and redoing kernel decomposition calculations. I believe there are native model grid versions available for the three radiative kernels. With the obtained difference, it may be easy to determine whether the error is acceptable. For NorESM2-MM with a low model top, the authors could even simplify the isolation process by using 17-level kernels for the kernel decomposition calculations. Actually, the sentence in lines 173-175 suggests the difference for kernels from radiative transfer codes is small.

We agree that this test would be useful to isolate how much different radiative transfer parameterisations impact the kernel results. However, we decided not to add these extra calculations to the manuscript to keep this section of the paper relatively concise with respect to our focus on the ozone experiments. As suggested, the reader can infer from the text that the difference between the magnitude of the stratospheric temperature adjustment is small.

Line 146: I was just wondering if the authors have any idea why there is a close agreement between cloud adjustment from the two different methods.

As noted above, we have changed our method for calculating the cloud adjustment. However, this still results in a very close agreement with the Smith et al. (2020) value but a more in-depth assessment of the two different methods would be needed to say more about why they agree so closely.

Lines 146-148: If I understood Smith et al. (2020a) correctly, the cloud adjustment in Smith et al. (2020a) was obtained by using APRP for SW and PRP for LW, and there is no liquid water path adjustment for the $CO_2$ cloud adjustment calculation. It would be great to double-check it.

This is correct, apologies for the misunderstanding. The text has now been updated (see lines 171-173):

*"In Smith et al. (2020a) shortwave and longwave $A_c$ are estimated separately using the APRP approach and offline monthly-mean partial radiative perturbation calculations, respectively."*

Lines 205-206 & 221-226: These sentences suggest a ~8K temperature difference at 10 hPa, and correspondingly, there is a 0.8 W m$^{-2}$ spread in the IRF. The resulting IRF sensitivity to temperature matches very well with the around -0.1 W m$^{-2}$ K$^{-1}$ (slopes) shown in Fig 1C and Fig S2 in He et al. (2023). As the online IRF difference (4 W m$^{-2}$) reported by He et al. (2023) includes both contributions from radiative transfer code difference and base-state difference, it may be better to compare the 0.8 W m$^{-2}$ spread here with the ~2 W m$^{-2}$ spread in offline IRF calculation with identical radiative transfer code (e.g., Fig 1B in He et al. (2023)).

This is a good point, thanks for highlighting this more suitable comparison. We have now amended the text (lines 261-269) to more usefully compare our results against those from He et al. (2023) Figure 1b and Figure 1c:

*"The IRF across all three experiments ranges from 4.9 – 5.7 W m$^{-2}$ resulting in a spread of 0.8 W m$^{-2}$. This is smaller than the spread of 2 W m$^{-2}$ (ranging from around 5 – 7 W m$^{-2}$) reported by He et al. (2023) for offline 'double-call' experiments of 4xCO$_2$ IRF calculated with a single radiative transfer code and base-states from the Atmospheric Model Intercomparison Project (AMIP) for 12 CMIP5/6 models (hence their spread is due only to differences in base-state). However, we find closer agreement between our IRF sensitivity to 10 hPa base-state temperature and He et al. (2023), who found a near -0.1 W m$^{-2}$ K$^{-1}$ relation between the spread in offline double-call experiments and air temperature at this level (Figure 1c in He et al., 2023). From Figure 2 it can be inferred that this matches very well with the ~7 K difference in base-state 10 hPa temperature between 'StratO$_3$x1.5' and 'StratO$_3$x0.5' and the corresponding 0.8 W m$^{-2}$ spread in IRF."*

Meanwhile, I hope the authors can discuss cloud adjustment more. Apparently, we can see the stratospheric temperature dependence in both IRF and SARF. The offsetting effects of cloud adjustment are the reason why the stratospheric temperature dependence does not extend to the ERF. It feels like the cloud adjustment probably mainly occurs for high clouds, which could be closely related to the response of tropopause to ozone perturbation. It would be good if the author could

decompose cloud adjustment into contributions from high, mixed, and low clouds, probably following Soden and Vecchi (2011). It would be even better if the authors had ISCCP simulator results archived. If the guess is correct, the authors could probably avoid the high cloud response (or tropopause response) by limiting the ozone perturbation within the upper stratosphere. In that case, the authors could probably get the stratospheric temperature-dependent ERF.

See earlier response (also below).

Lines 254-255: It is because of the offsetting effects of cloud adjustment.

As shown in the updated version of Figure 3, the magnitude of the cloud adjustment in each experiment is now much more comparable and so this adjustment does not offset the IRF as previously implied. The cloud adjustment in the 'Strat $O_3$x0.5' case now decreases by 0.14 W m$^{-2}$ (from 1.68 W m$^{-2}$ in the 'Standard' case), and whilst this does appear to offset the increased IRF (and the very slightly increased stratospheric adjustment), the upper bound of standard error falls within the lower bound of the 'Standard' and '$O_3$x1.5' case, making it a lot more uncertain to state the offsetting role of clouds.

Lines 255-256: It could be expected, considering the similarities between CESM2 and NorESM2.

We agree with this comment and a similar point that was raised by Reviewer 1. We have now deleted this comparison from the text and supplementary, seeing as the comparison was somewhat limited and further calculations would now need to be performed (for the CESM2 cloud adjustment).

Lines 263-265: I wondered why the authors expect a large spread in the magnitude of stratospheric temperature adjustment. It looks like Fig S6 in He et al. (2023) shows almost no difference in the stratospheric adjustments obtained from piClim-4xCO2/piClim-control and amip-4xCO2/amip simulations.

This is a useful result from He et al. (2023), and one we had not seen prior to running these experiments. We now refer to this result in the text (lines 257-260):

*"This corroborates experiments from He et al., (2023; Figure S6) that compare the size of stratospheric temperature adjustment after a quadrupling of $CO_2$ from two different base-states; the multi-model ensemble-mean difference in their adjustment is just -0.03 W m$^{-2}$ (although the model range is considerably larger at around 0.5 W m$^{-2}$)."*

Due to the strong influence of ozone on stratospheric temperature we anticipated to see more of an effect of our idealised experiments on the ERF via both the impact on IRF and through the impact stratospheric adjustment, given that stratospheric cooling is the dominant adjustment for $CO_2$.

---

## Author Comment (AC3)

We thank Reviewer 1 for their positive and helpful comments towards improving this manuscript. Our responses are given in the blue font. Where necessary, we refer to the revised version of the manuscript.

*General comments:*

Byrom et al. have created a clear manuscript that conveys scientifically relevant findings. They do this by expanding upon work that has highlighted a stratospheric base state dependence on CMIP6 model 4xCO2 ERF, investigating the role of O3 that plays a significant role in determining stratospheric temperatures.

They construct stratospheric ozone change experiments in NorESM2, prescribing ozone from the higher stratospheric-resolution CESM2-WACCM, to demonstrate how stratospheric temperatures change with O3 concentrations. They then calculate IRF (through offline radiative code) and adjustments (through radiative kernels) that result from a quadrupling of CO2 from these different stratospheric states.

Through this, and by comparing methods with previous work, they show the importance of calculating IRF through offline code, and using appropriate kernels that avoid extrapolating to above the top of the model. They further show that IRF depends on the base state of the stratosphere, due to overlapping O3 and CO2 spectral bands and stratospheric temperature. However, stratospheric temperature adjustments do not change significantly between increases and decreases of O3. Cloud adjustments result in invariant ERF, which highlights that stratospheric O3 concentrations are unlikely to explain the inter-model spread in 4xCO2 ERF.

I would like to see this published, though I have some minor concerns that I would like to see addressed.

*Specific comments:*

L112: on calculating the cloud adjustment as a residual, I'm not confident that A_c is being properly estimated. It seems like the same criticism levied against calculating IRF as the residual could apply to calculating A_c as a residual. Non-zero $\epsilon$ may explain the differences in cloud feedbacks shown in Figure 3. I'd like to see further justification for assuming approximately zero $\epsilon$.

We have now calculated the cloud adjustment following the adjusted cloud radiative effect method of Soden et al. (2008; see https://journals.ametsoc.org/view/journals/clim/21/14/2007jcli2110.1.xml), whereby the cloud radiative effect is calculated as the difference between clear-sky and all-sky ERF and adjusted to correct for cloud masking of the clear-sky forcing and adjustments. For Figure 1 we use the NorESM2-MM all-sky and clear-sky $4xCO_2$ shortwave and longwave ERFs and correct for the cloud masking by using output from the corresponding PORT 4xCO2 simulation and clear-sky and all-sky radiative adjustments from the CESM-CAM5 kernels:

dLW_cloud = -d_cre_lw + cloud_masking_of_forcing_lw + (dLW_q_cs - dLW_q) + (dLW_ta_cs - dLW_ta) + (dLW_ts_cs - dLW_ts)

 dSW_cloud = -d_cre_sw + cloud_masking_of_forcing_sw + (dSW_q_cs - dSW_q) + (dSW_alb_cs - dSW_alb)

This calculation is repeated for the cloud adjustment in Figure 3 using the data from the respective experiments presented.

Note that Øivind Hodnebrog provided calculations for this cloud adjustment, hence his addition as a co-author on the manuscript.

L117: It's not clear to me how you calculate the adjustment to surface temperature change (A_T_s). I think a brief explanation is warranted, or just clarification that you used the same method as in the papers that follow the same approach.

We use radiative kernels to calculate this adjustment in the same way that all of the other adjustments are calculated (apart from the cloud adjustment). With regard to the sentence: "Several studies follow this approach (e.g., Hansen et al., 2005; Forster et al., 2016; Smith et al., 2018)." – this was intended to highlight that several other studies also calculate fixed SST ERF in the same way (whereby land-surface temperatures are allowed to respond to the forcing given the difficulty in prescribing fixed surface temperatures) and not necessarily that we calculate (A_T_s) in the same way as these papers (although we do with respect to Smith et al. 2018). This section of the text has now been updated to be clearer (lines 132-137):

*"Additionally, we also use kernels to calculate the adjustment due to surface temperature change ($A_{T_S}$) as in Smith et al. (2018), since land surface temperatures are allowed to respond to the forcing in our simulations given the difficulty in prescribing fixed surface temperatures (Forster et al., 2016). Several studies also follow this approach whereby the calculation of ERF includes the radiative response of land surface warming or cooling (e.g., Hansen et al., 2005; Forster et al., 2016; Smith et al., 2018; Smith et al., 2020a)."*

Figure 3: especially given the small number of years (9, instead of the ~20 you might use from 30 year runs) used in calculating values in the figure, this figure could benefit from error bars to show standard errors. If error bars are very small, I think the results would still benefit from acknowledging this.

We agree and we have now added standard error bars to this figure (see revised manuscript).

Fig. S3: CESM2 and NorESM2 are very similar, so isn't comparing what they do here quite limiting? It might be worth clarifying this.

Yes, it is quite limiting and the use of an altogether different model with a different radiation scheme would definitely be interesting to use. NorESM2 uses a different module for aerosol physics and chemistry, including cloud and radiation interactions and has further differences in the land component and surface albedo calculation. However, we have now deleted this comparison from the text and supplementary seeing as the comparison was somewhat limited and further calculations would now need to be performed (for the CESM2 cloud adjustment).

*Technical corrections:*

L17-19: Unclear sentence "However… O3". This could read as though the base-state stratospheric temperatures and the spectral overlap of CO2 and O3 are counteracting the spread in CO2 ERF. Given what you write in your conclusions, it seems like this should be something along the lines of "The spread in CO2 IRF is explained by the impact of base-state stratospheric temperature on the emission of outgoing longwave radiation and the spectral overlap of CO2 and O3, but these do not explain the spread in CO2 ERF".

Thanks for pointing this out, we have now amended the text to be clearer as suggested (lines 16-19):

*"These experiments impact the IRF due to the influence of base-state stratospheric temperature on the emission of outgoing longwave radiation and the spectral overlap of $CO_2$ and $O_3$. However, the impact on IRF does not result in a correspondingly large spread in $CO_2$ ERF. We conclude that inter-model differences in stratospheric $O_3$ concentration are therefore not predominantly responsible for inter-model spread in $CO_2$ ERF."*

L233-244: I think the use of brackets here leads to less clarity. This seems like a good example of what has been criticised in previous literature (https://doi.org/10.1029/2010EO450004), where parentheses are not used for clarification. I would recommend writing separate sentences e.g. for increase vs. decrease of O3 instead of trying to save space with parentheses.

This has been noted and the sentence has now been re-written as two separate sentences (see lines 277-289):

*"As stratospheric $O_3$ concentrations increase, TIR absorption at these wavelengths also increases to an extent that depends on the level of band saturation and the abundance of other gases absorbing at these wavelengths. The opposite occurs if stratospheric $O_3$ concentrations decrease. For $CO_2$, the main TIR bands lie in the window regions of the $H_2O$ spectrum, with absorption centered at 4.3 µm and 15 µm (the latter of which is highly significant due to its proximity to the peak of blackbody distribution for the Earth's effective emitting temperature). Weaker bands also occur near 10 µm. Regions of spectral overlap between $O_3$ and $CO_2$ therefore arise at several wavelengths; at 15 µm the strength of $CO_2$ absorption largely masks the radiative effect of $O_3$ at 14.27 µm and absorption by both gases at 4.75 µm and 4.3 µm has little impact given their location further away from the peak of Earth's blackbody distribution. However, decreased stratospheric $O_3$ concentration leads to weakened absorption at 9.6 µm that can enhance absorption by $CO_2$ at 10 µm. Likewise, increasing stratospheric $O_3$ concentration results in strengthened 9.6 µm absorption that mutes $CO_2$. Combining the effect of base-state stratospheric temperature and spectral overlap, a 4x$CO_2$ perturbation therefore results in an enhancement of the IRF in the 'Strat $O_3$x0.5' case relative to the 'standard' experiment. Correspondingly, an increase in stratospheric $O_3$ has the opposite (albeit evidently weaker) effect."*

Figures (generally): the figures are a bit blurry, which makes it hard to read some elements especially in Figs. 1, 3, and S3. Ideally these would be higher resolution, or the font size might be increased.

The figures have now been updated using a higher resolution, please see the revised manuscript. Apologies for the blurriness.

Fig. S3: why the change of colours from Fig. 3? I found it a bit harder to distinguish the shades of blue than the shades of orange.

Originally the results from NorESM2 and CESM2 were plotted together, so the orange and blue shades were used to distinguish the results from each model. But the CESM2 comparison has now been deleted from the manuscript.

---

## Author Comment (AC4)

We thank Reviewer 3 for their positive and careful review. As with Reviewer 1 and 2, we appreciate the attention that has been given to our manuscript, particularly regarding their comments related to the cloud adjustment and the comparison with He et al., (2023). Our responses are given in the blue font. Where necessary, we refer to the revised version of the manuscript.

Recent studies have argued that a large component of the CO2 IRF spread in CMIP models can be explained by documented large spread in stratospheric base state temperatures. This manuscripts serves as an important follow-up, testing whether differences in stratospheric O3 can explain the documented stratospheric temperature spread and thus the CO2 forcing spread. The authors find the answer is no. It's a well written, interesting study. But before it can be accepted I recommend the authors address my comments below that touch on interpretation of results and providing more details/explanations in certain places.

General: I agree that Stratospheric O3 differences cannot explain the spread in 4xCO2 ERF, based on the results presented here. But I think the authors are too quick to discount the effect of differing Stratospheric O3 on the spread in IRF. As noted a few comments below, I'd argue that a slightly deeper dive into the He et al. Results suggests the two papers may be more comparable than the authors suggest.

Introduction Section: To address the relevance of this study's results to the question of CMIP spread, it would be helpful answer how realistic is the range of 0.5 StratO3 to 1.5 StratO3 relative to the actual range of StratO3 across CMIP models. Is there a considerable spread in Strat O3 across CMIP models? I was under the (maybe incorrect) impression that O3 is prescribed in CMIP atmosphere-only simulations. In which case, the hypothesis that Strat O3 spread explains the CMIP Strat. T spread would be wrong. Some comments/explanation along these lines would be helpful.

We have now added another paragraph to the introduction/methods to better frame our motivation related to the spread in CMIP6 models (lines 66-73 and 91-93).

*"One such cause could relate to stratospheric $O_3$ – a key constituent in modulating stratospheric temperature. Depending on the treatment of stratospheric chemistry, models adopt a range of methods to generate $O_3$ fields using either an interactive chemistry scheme, a simplified online scheme or a prescribed pre-simulated dataset. Consequently, the resulting spatial structure and regional distribution of concentrations can differ substantially. Keeble et al. (2021) evaluate long-term $O_3$ trends in 22 CMIP6 models and find poor agreement in the simulation of pre-industrial total column ozone (TCO), with a variation from 275 to 340 DU between 60˚N-60˚S. Further, a ~ 20 DU range is observed between 10 of the models that prescribe stratospheric $O_3$ according to the CMIP6 $O_3$ dataset (Checa-Garcia, 2018), highlighting that even the model-specific implementation of common input can lead to significant differences in TCO."*

*"Considering the substantial range in pre-industrial TCO noted by Keeble et al. (2021, Figure A3), we choose such a large, idealised increase/decrease in attempt to cover a broader range of stratospheric $O_3$ than shown by CMIP6 models, thus any effect on 4xCO$_2$ ERF would likely be amplified in comparison."*

Here we refer to the Keeble et al., (2021 https://acp.copernicus.org/articles/21/5015/2021/) study that evaluated long term historical ozone trends in 22 CMIP6 models (6 of which used interactive chemistry schemes, 3 use a simplified scheme, 10 use the prescribed CMIP6 ozone dataset and 3 use prescribed ozone values from CESM-WACCM simulations). Despite good agreement between the CMIP6 multi-model mean and observations, this study shows substantial variation between individual models (see Keeble et al., (2021) Figures A1 and A2). Across the 2000-2014 period they report that notable differences occur in the uppermost stratosphere and around the tropopause, and across 1960-

2014 they show that ozone fields are both overestimated and underestimated by models that use interactive chemistry. They further report poor agreement in the simulation of pre-industrial TCO, varying between 275 and 340 DU (see their Figure A3) and furthermore that there is a ~ 20 DU range in pre-industrial TCO between the 10 models that prescribe ozone using the same CMIP6 dataset. 11 of the models analysed by Keeble et al. (2021) are used in the Smith et al. (2020) study (in Keeble et al. 7 of these models use prescribed ozone, 3 use an interactive chemistry scheme and 1 uses a simplified online scheme). However, even if all of the Smith et al. models used prescribed $O_3$ in their atmosphere-only set-up (i.e., piClim-control), we conclude that there could still be spread in stratospheric $O_3$ based on Keeble et al's. findings. Although Morgenstern et al. (2020; Figure S1 https://doi.org/10.1029/2020GL088295) actually demonstrate that there is a large spread in TCO between models that use interactive chemistry in piClim-control simulations (including GFDL-ESM4, MRI-ESM2-0 and UKESM-0-LL which are used in Smith et al., (2020), see their Table 1). Thus, we further conclude that at least some of the atmosphere-only simulations do include full ozone chemistry.

Line 80-90. Although it's clear if you read table S1, I recommend the authors make it clearer in this section of the text that stratospheric O3 is reduced (or increased) by 50% in both the perturbed 4xCO2 simulation and in the corresponding control pre-industrial simulation, thereby ensuring the new O3 filds act only to alter the base state and not as a forcing adjustments itself.

Thanks for this point, we appreciate that this will make our experimental design clearer to the readers and we have the following sentence (lines 94-95): "*Note that in each ERF experiment the $O_3$ increase (or decrease) is applied to both the control and $4xCO_2$ simulation so that the new $O_3$ field acts exclusively to alter the base-state atmosphere and does not act as a forcing itself.*"

Line 150-190: I appreciated the authors nuanced discussion about the application of radiative kernels for diagnosing the stratospheric adjustment. Their arguments logically make sense. But do we have any proof that their estimate of the stratospheric adjustment using the CESM kernel is more representative of the model's true adjustment compared to the previous uses of kernels applied in e.g. Smith et al.?  The fact that the residual-derived cloud adjustment matches the alternative Smith et al method is maybe promising, but as a residual calculation, it is difficult to pinpoint potential canceling biases. For instance, it's possible both the stratospheric adjustment and some other adjustments have equal and opposite errors.

We think that our estimate of the stratospheric adjustment is more representative of our NorESM2 experiment's adjustment because we do not extrapolate our data to levels beyond the uppermost atmospheric level, and so do not include the adjustment in temperature that occurs beyond these pressure levels (as done by Smith et al. 2020). Also, to a lesser extent, because the CESM-CAM5 kernel is built using the same radiative transfer code as NorESM2 i.e., RRTMG. With regard to the cloud adjustment, please see our response below.

If we assume the authors are correct in their statement that it is best to use kernels from the host model, it would be helpful if they also gave a recommendation about how kernels should best be used when being applied across multiple models to evaluate inter-model spread.

Although its not a focus of this paper, a brief comment would be helpful since this is a common use of kernels and there is not currently a radiative kernel available for every host model.

We appreciate this point, and it could be an option to recommend not extrapolating model data to kernel levels that are outside of the model's native bounds (see lines 216-219):

*"If a radiative kernel is not available for a given model, or a kernel needs to be applied across multiple models to evaluate inter-model spread, then it could be more suitable to not extrapolate data outside of each model's native vertical bounds. However, the best use of kernels is likely quite case specific."*

Line 220-226: First, the authors state that their range in IRF across experiments of 0.8 W/m2 is much smaller than the 4 W/m2 IRF spread that He et al. Finds across the online double calls. This is true. But the authors should keep in mind that He et al. Does not claim all of that spread is due to the base state. They claim "more than half" (presumably the r^2 ~ 0.67 is what they are basing this on) of the spread is explained by the base state but not all of it. I recommend the authors factor this in when comparing the He et al. result to their own findings.

Further, I'd argue that a fairer comparison between the spread in this paper and the spread in He et al. would be a comparison to the offline calculations of their figure 1C (rather than their 1B) where the IRF spread is subject only to base state differences and not to differences in radiative transfer algorithms across models, as is the case in this present study. In the He et al. figure 1C it appears ~14 degreesK of 10 hPa stratospheric temperature spread across models corresponds to 1.3 W/m2 of IRF spread. Since the 10 hPa temperatures in this study range from -3K to +4K relative to the standard case (line 205), and this corresponds to a 0.8 W/m2 spread in IRF across the experiments, it would appear the StratT vs IRF spread results are actually quite comparable between the two studies from this perspective. Does this impact the overall conclusions that the authors would draw about the importance of StratO3 spread to IRF spread?

Reviewer 2 also raised this point and we agree that the comparison to Figure 1c of He et al. (2023) provides a better evaluation of our results against theirs. We now compare against the near 2 W m$^{-2}$ spread in IRF from Figure 1b of He et al. (2023) and then discuss the similarity between our change in temperature at 10 hPa (between the 50% increase and decrease base-state) and the spread in IRF and Figure 1c from He et al. (2023) (see lines 261-269). Our conclusion remains the same with respect to the impact of stratospheric O$_3$ increases/decrease on 4xCO$_2$ IRF, i.e., that our idealised experiments show a dominant impact on the IRF magnitude.

Line 231-244: It is interesting to consider the relative importance of StratO3 effect on CO2 forcing through overlap with CO2 vs through stratospheric temperature effects. Do the authors have a relative sense of this? It's difficult to imagine a setup that could address this for ERF, but for IRF one could presumably perform an offline radiative transfer calculation with PORT where StratO3_x0.5 or StratO3_x1.5 is imposed but Stratospheric temperatures are prescribed in all cases from a StratO3_x1 climate as a way to isolate the spectral overlap effects from the stratospheric temperature base state secondary effects.

We previously performed LBL tests (using the GENLN2 code) to compare the relative importance of these effects. We found that a 50% reduction in O$_3$ leads to a +0.07 W m$^{-2}$ increase in 4xCO$_2$ IRF (from 4.2 W m$^{-2}$), whilst a reduction in the temperature of -2K across the whole stratosphere leads to a 0.16 Wm$^{-2}$ increase. Based on these results the effect of stratospheric temperature dependence is stronger than the effect of spectral overlap. Furthermore, as shown in Figure 2, decreasing stratospheric ozone by 50% results in widespread cooling of the stratosphere with ΔT values largely more negative than -3 K and peaking at -9 K. We have now added these results as a footnote on page 12 of the revised manuscript:

*"Tests performed by the GENLN2 line-by-line (Myhre et al., 2006) show that a decrease in temperature of 2 K across the whole stratosphere leads to a 0.16 W m$^{-2}$ increase in 4xCO$_2$ IRF, whilst a 50% reduction in stratospheric O$_3$ leads to a 0.07 W m$^{-2}$ increase in 4xCO$_2$ IRF."*

Line 253-255: There appears to be a misinterpretation here. The fact that A_Tstrat remains the same size across experiments would actually support the IRF enhancement/reduction extending all the way to ERF rather than prevent it (As ERF = IRF + A_Tstrat + other adjustments). In order for the IRF enhancement/reduction not to extend to ERF, there needs to be an equal but opposite compensation in the enhancement/reduction of a different adjustment. In Figure 3, this seems to occur largely through the cloud adjustment term. i.e. the IRF is larger than the standard experiment for the StratO3x0.5 case while the A_c is smaller than standard. Likewise the IRF is smaller than the standard experiment for StratO3x1.5 while the corresponding A_c is larger than standard. I recommend the authors rephrase this section to emphasize the A_c term changes rather than focusing on the static magnitude of A_Tstrat. I further recommend the authors explore why A_c has this apparent sensitivty to StratO3. It would help us understand whether the ERFs lack of sensitivity is due to the specific characteristics of stratospheric O3 or if ERF is just not sensitive to stratospheric temperature base states more generally.

This is a great point and we now realise our apparent oversight of the possible importance of the effect on the cloud adjustment.

However, following this comment (and similar points from Reviewer 1 and 2) we decided to change our method for calculating the cloud adjustment to the adjusted cloud radiative effect method of Soden et al. (2008; https://journals.ametsoc.org/view/journals/clim/21/14/2007jcli2110.1.xml). As shown in the updated version of Figure 3, the magnitude of the cloud adjustment in each experiment is now much more similar, demonstrating that this adjustment doesn't offset the impact of ozone increase/decreases on IRF and SARF as implied previously.

---

## Author Response (AR2)

We thank the reviewer for their additional comments on this manuscript. Please find our responses given in the blue font below (note all references to line numbers refer to the track changes version of the revised manuscript).

Lines 98-99: Given that the O3 perturbation is bounded by the approximated tropopause [i.e., the linearly varying tropopause (from 100 hPa at the equator to 300 hPa at the poles)], does this imply that there are O3 perturbations within the actual troposphere? Alternatively, does this tropopause setting lead to O3 changes within the upper troposphere and subsequently affect high clouds? It would be helpful to provide the global-mean or zonal-mean vertical cloud fraction climatology distributions for the different control experiments (standard, 0.5x, and 1.5x Ozone) and the corresponding cloud fraction changes to 4xCO2. Additionally, it would be beneficial to include both the all-sky and clear-sky 4xCO2 IRF to evaluate whether the cloud masking value changes with different O3 settings.

We have now included zonal-mean cloud fraction distributions for each case (base-state and 4xCO$_2$) in supplementary Figure S3 and these are referred to in the main text (see lines 313-316). From these plots we see that the ozone experiments do lead to changes in high cloud fraction (decreasing in 'Strat O$_3$x1.5' and increasing in 'Strat O$_3$x0.5'). Though this appears to occur largely below the approximated tropopause level and the increase/decrease in ozone (see Figure S3a). We have also included all-sky and clear-sky (SW and LW) IRF values in Table S3 of the supplementary and we (briefly) refer to the cloud-masking effect of each experiment on lines 316-320. We hope that the inclusion of more cloud details now helps the reader to understand the role that cloud changes play on radiative fluxes in each experiment.

Lines 107-108: Can the authors clarify what is meant by "12-month climatology"? Is this a multi-year mean climatology (e.g., 25 or 30 years), or does it specifically refer to "the first 12 months of its output," as mentioned at the end of the sentence?

This referred specifically to the first 12 months of output as mentioned at the end of the sentence, but the text has now been re-worded to avoid any confusion. Hopefully this is now clearer. It now reads as:

*"It is used here to perform two sets of radiative transfer calculations for each experiment listed in Table S1; a baseline (control) simulation and a perturbed (4xCO$_2$) simulation, which are both run using a year's worth of climatology from the corresponding ERF control integration (i.e., the first 12 months of its output)."*

Lines 127-129: Given the definition of ERF as the net downward radiative flux difference, the cloud radiative effect change (dCRE) should correspond to the difference between all-sky and clear-sky ERF. As such, there should be no negative sign preceding d_cre_lw and d_cre_sw in the adjusted cloud radiative response section (as noted in response to Reviewer #1).

Thanks for highlighting this, we agree there should be no negative sign before d_cre_lw and d_cre_sw, apologies for not spotting this in the response to Reviewer 1. Looking back at our python script, I see that we initially calculate LW CRE (d_cre_lw) and SW CRE (d_cre_sw) as clear-sky minus all-sky, and the minus sign was added later in the script to correct this as all-sky minus clear-sky when coding the equation for dLW_cloud and dSW_cloud. These equations were then copied in the response to Reviewer 1.

Lines 129-132, 164-169, and 241-242: Since the radiative kernel method is now being used for cloud adjustment calculations, I recommend rephrasing these sentences for clarity. Additionally, because the revised method has resulted in non-closure in the decomposition, it would be helpful to provide the residual term values for all decomposition plots in the manuscript. While the original radiative kernel method, the method used by Smith et al. (2020) (including APRP and PRP for cloud adjustment), and the previous version of this manuscript had no residual terms, residuals are apparent in the current version. Furthermore, could the authors provide comparisons of dCRE, cloud masking of IRF, and cloud masking of other adjustments for these 4xCO2 simulations? It would be good to understand why the cloud adjustment values are different between previous and current version of decompositions.

We have now updated the text, thanks for highlighting this oversight (see lines 130-135). We have also included the residual term in Figure 1 and Figure 3 and made reference to this in the text (see lines 322-325).

We have also provided comparison of dCRE and the cloud masking of IRF and adjustments as figures in the supplementary and we refer to these plots in the main text (see lines 309-320). Now that we calculate the cloud adjustment with the use of kernels, residual term $\epsilon$ is not aliased into our values and we can compare the relative contribution of dCRE and cloud masking in each experiment.

Lines 132-138: Do the ERF values reported in the manuscript represent land-warming-corrected ERF?

No, the ERF values aren't corrected for land surface warming and so don't represent land-warming corrected ERF. We have now added the following text on lines 141-143 to clarify this:

"Subtracting $A_{T_S}$ from the ERF could provide a land surface warming corrected forcing (following Smith et al., 2020a), however we do not calculate this here. Instead, we report the magnitude of $A_{T_S}$ to inspect any change in its value with each $O_3$ experiment."

Figure Caption of Figure 1: Note that Smith et al. (2020a) calculated non-cloud adjustments using the radiative kernel method and cloud adjustments using APRP and PRP for SW and LW, respectively.

Thanks for pointing out that this needed to be clarified in the caption. The caption text has now been updated.

Lines 181-182: The statement "whereby geopotential height is used as an approximation of geometric height on model pressure levels in the control integration" seems unnecessary and could be omitted for conciseness.

We agree, this has now been deleted from the text.

Lines 299-300 and 309-311: It would be helpful to explain why stratospheric O3 changes do not significantly affect the magnitude of ERF, despite the significant differences shown in the IRF. Is there a specific adjustment or residual term offsetting the IRF's contribution? If not, why does the ERF fail to reflect the difference observed in the IRF and SARF?

We agree that this would be helpful and we have added text to discuss these points (lines 302-325). Here, we specifically highlight the strengthening/offsetting impact of the adjustments and the residual

term to explain to the reader why the ERF fails to reflect the differences seen in the IRF. We also further discuss the stratospheric temperature adjustment (see lines 260-270) to give the reader more background on the nature of this important adjustment.

Lines 311-314: Given the secondary contribution from the spectral overlap of CO2 and O3 to the magnitude of IRF, as demonstrated in the revision, could the authors highlight the dominant role of stratospheric temperature dependence? The current phrasing might imply that the two mechanisms contribute comparably.

We have now clarified this in the conclusion (see lines 347-349).

*"Instead, these experiments demonstrate a dominant impact on the magnitude of IRF, primarily due to the impact on base-state stratospheric temperature with an ancillary effect from spectral overlap of $CO_2$ and $O_3$".*

And also in the abstract (see lines 16-18).

"These experiments impact the IRF primarily due to the influence of base-state stratospheric temperature on the emission of outgoing longwave radiation, with the spectral overlap of CO2 and O3 playing a subsidiary role."

General Comment: Could the authors clarify why the albedo and water vapor adjustment values differ by eye between the previous and current versions in the tracked document?

Apologies for not having highlighted this in our previous revision. The albedo adjustment values differ because we originally opted to use the 'asdir' and 'asdif' (i.e. direct and diffuse surface albedo) NorESM2 output fields in our calculation (which are used in the construction of the albedo kernel, see: https://essd.copernicus.org/articles/10/317/2018/). However, we realised that the user is meant to use the shortwave surface radiation fields to calculate surface albedo (FSDS and FSNS), so we updated our calculations accordingly. The $H_2O$ adjustment differs for a similar reason, whereby we updated our method to be fully consistent with Pendergrass et al. (2018, https://github.com/apendergrass/cam5-kernels/tree/master). A lot has been learnt throughout this paper regarding the use of kernels, apologies for unintentionally excluding this explanation before.